Anatomical mechanism of spontaneous recovery in regions caudal to thoracic spinal cord injury lesions in rats

Li Lu-sheng 1 3
Yu Hao 1
Raynald Raynald 2
Wang Xiao-dong 2
Dai Guang-hui 2
Cheng Hong-bin 2
Liu Xue-bin 2
An Yi-hua anyihua_wj@sina.com 1 2
1 Department of Neurosurgery, Beijing Sanbo Brain Hospital, Capital Medical University , Beijing , China
2 Department of Functional Neurosurgery and Cytotherapy, General Hospital of Chinese People’s Armed Police Forces , Beijing , China
3 Department of Neurosurgery, Beijing Chao-yang Hospital Affiliated with Capital Medical University , China
Abdullah Jafri
Electronic publication date: 2017 Jan 10
Publication date: 2017
Volume: 5
Electronic Location ID: e2865
Received 2016 Oct 27; Accepted 2016 Dec 5
Copyright: ©2017 Li et al.
Copyright year: 2017
Copyright holder: Li et al.
License: This is an open access article distributed under the terms of the Creative Commons Attribution License, which permits unrestricted use, distribution, reproduction and adaptation in any medium and for any purpose provided that it is properly attributed. For attribution, the original author(s), title, publication source (PeerJ) and either DOI or URL of the article must be cited.
License URL: https://creativecommons.org/licenses/by/4.0/

Keywords: Spinal cord injuries, Transection, Electrophysiology, Repair, Function recovery, Hindlimbs, Nerve regeneration

Funding: Beijing National Science Foundation 7092017 Core Technology Research Projects of Strategic Emerging Industries in Guangdong Province 2011A081401003 This research was supported by the Beijing National Science Foundation (7092017), the Core Technology Research Projects of Strategic Emerging Industries in Guangdong Province (2011A081401003). The funders had no role in study design, data collection and analysis, decision to publish, or preparation of the manuscript.

==============================
Background

The nerve fibre circuits around a lesion play a major role in the spontaneous recovery process after spinal cord hemisection in rats. The aim of the present study was to answer the following question: in the re-control process, do all spinal cord nerves below the lesion site participate, or do the spinal cord nerves of only one vertebral segment have a role in repair?

Methods

First we made a T7 spinal cord hemisection in 50 rats. Eight weeks later, they were divided into three groups based on distinct second operations at T7: ipsilateral hemisection operation, contralateral hemisection, or transection. We then tested recovery of hindlimbs for another eight weeks. The first step was to confirm the lesion had role or not in the spontaneous recovery process. Secondly, we performed T7 spinal cord hemisections in 125 rats. Eight weeks later, we performed a second single hemisection on the ipsilateral side at T8–T12 and then tested hindlimb recovery for another six weeks.

Results

In the first part, the Basso, Beattie, Bresnahan (BBB) scores and the electrophysiology tests of both hindlimbs weren’t significantly different after the second hemisection of the ipsilateral side. In the second part, the closer the second hemisection was to T12, the more substantial the resulting impairment in BBB score tests and prolonged latency periods.

Conclusions

The nerve regeneration from the lesion area after hemisection has no effect on spontaneous recovery of the spinal cord. Repair is carried out by all vertebrae caudal and ipsilateral to the lesion, with T12 being most important.

Introduction

The brain is plastic, and mammals are capable of spontaneous recovery after spinal cord injury. The mechanisms underlying this process are not yet clear and are disputed. After brain or spinal cord injury, many researchers have found that transplantation of multifunctional three-dimensional scaffolds and stem cells treatment with neurotrophic factors, administration of small molecules, or genetic modifications in the lesion area promote neuronal regeneration in the lesion and improve motor function recovery (Estrada et al., 2014; Fan et al., 2010; Jee et al., 2012; McCall, Weidner & Blesch, 2012; Shi et al., 2015; Tan et al., 2016; Wright et al., 2011). In addition, several groups have shown that, after hemisection of the thoracic spinal cord, both hindlimbs show significant improvement 3–5 weeks later. Moreover, if a second hemisection on the side contralateral to the lesion was performed, rats showed complete paralysis of both hind limbs with no signs of recovery of locomotor function over four weeks. For example, Courtine et al. (2008) first produced a left–side hemisection model at thoracic segment 12 (T12 refers to the spinal cord level). Then, 10 weeks later, they performed a second hemisection on the contralateral side (T7). The results showed that the rats initially lost all movement on the T7 side and some of the movement on the T12 side. However, when the rats were subjected to the T7 and T12 hemisection at the same time, both hindlimbs instantly lost all movement (Courtine et al., 2008). This observation demonstrates that nerve fibers around the lesion participate in repair. These fibers must originate rostral to the lesion on the ipsilateral side, then cross the midline to the contralateral side, travel down the spinal cord, and re-cross the midline caudal to the lesion (Ballermann & Fouad, 2006; Courtine et al., 2008; Etlin et al., 2010; Reed, Shum-Siu & Magnuson, 2008). These two different repair mechanisms are in conflict with each other, and further research is needed to confirm and explain the repair process. If the mechanisms of repair in both contexts were resolved, it would inform novel treatments to promote recovery and further improve limb function in patients with spinal cord injury.

Among the questions that remain to be answered is whether the nerve fiber circuits that control the ipsilateral hindlimb after injury are comprised of spinal cord nerves in a single vertebra or spinal cord nerves in multiple vertebrae contribute to these nerve fiber circuits and participate in the recovery process. Accordingly, we designed a study to resolve these two possibilities. We first carried out a hemisection of thoracic vertebrae 7 (T7) on the left side of spinal cord, then eight weeks later performed a second hemisection of thoracic vertebrae 8–12 (T8–T12) on the ipsilateral side We then compared the extent of recovery of hindlimb function acrosss each hemisection group. In this study, abbreviations such as T7 indicate the vertebral segments.

Materials and Methods

Animals

In this study, we used adult Sprague–Dawley rats (200–220 g). All rats were allowed to acclimate to the new environment for seven days before the start of any experimental procedures. The rats were housed on a 12 h light-dark cycle with food and water provided ad libitum. All rats were deeply anesthetized before any surgical procedures were performed (10% chloral hydrate, 35 ml/kg). After surgery, antibiotics (Penicillin, 128,000 UI/kg) and 10 ml of sterile saline were administered subcutaneously each day during the first week. After each operation, the rat was placed in a separate cage for seven days before it was housed with other rats. The rats in the study were obtained from the Vital River Company. Ethical approval was obtained from the Beijing Neurosurgical Institute Laboratory Animals Ethics Committee in China.

In the first part of study, there were 50 rats in total and 10 rats died throughout the study; the mortality was 20%. In the second part of study, there were 125 rats in total and 16 rats died throughout the study; the mortality was 12.8%.

Groups

In the first part of the study, we investigated if regenerated axons coursed ipsilaterally or contralaterally. All rats underwent two operations on T7 spinal cord and were divided into 5 groups (Figs. 1A–1E). The first group was a sham operation group (N = 10) in which both operations were sham. Groups 2–5 all first received a hemisection of the left side of spinal cord at T7, but had different second operations. The second group was the control group (N = 6), in which the second operation was sham, consisted a midline cut in the spinal cord at T7. The third group was the ipsilateral experimental group (N = 8), in which a second hemisection was conducted on the ipsilateral side. The fourth group was the contralateral group (N = 8) in which a second hemisection was made on the contralateral side. The fifth was the transection group (N = 8), which received a full transection at T7 in the second operation.

Figure 1 Demonstrations of the operations for each group in first part of this study.

All the operations were made at T7. A black arrow indicates the level of the first hemisection operation; a red arrow indicates the level of the second hemisection operation (A) The sham group underwent two sham operations. (B) The control group first underwent a hemisection and a subsequent sham operation. (C) The ipsilateral group first underwent a hemisection and then a second hemisection operation on the ipsilateral side. (D) The contralateral group first underwent a hemisection operation and then a second hemisection operation on the contralateral side. (E) The transection group first underwent hemisection operation, which was followed by a transection operation.

In the second part of the study, we investigated the innervation of vertebrae downstream of the lesion site to determine if regenerated axons target single or multiple vertebrae. These experiments comprised three main groups, each of which were divided into five smaller sub-groups defined according to the site of the second operation: T8–T12. The first group was the operation group (N = 37, Figs. 2A–2E), in which all rats underwent 2 hemisection operations. The first operation was at T7 on the left side; eight weeks later the second operation was carried out ipsilaterally at a single vertebra from T8–T12 depending on sub-group. The second group was the control group (N = 37, Figs. 2F–2J). All received a hemisection at T7 on the left side, and eight weeks later underwent a sham operation at a single vertebra at T8–T12 depending on sub-group. The third was the sham operation group (N = 35, Figs. 2K–2O), in which rats in all sub-groups received a sham operation at both time points.

Figure 2 Demonstrations of the operations for each group in second part of this study.

A black arrow indicates the level of the first hemisection operation; a red arrow indicates the level of the second hemisection operation. Operations in second part of this study. A black arrow indicates the level of the first hemisection operation; a red arrow indicates the level of the second hemisection operation. (A–E) T8–T12 sub-groups in the operation group. (F–J) T8–T12 sub-groups in the control group. (K–O) T8–T12 sub-groups in the sham group.

Hemisection operation

The first hemisection operation

We generated the first hemisection operation according to previously published methods (Arvanian et al., 2009). In the operation, we used sharp scalpel not scissors to separate along the midline of spinal cord, which might cause less injury to the spinal cord. Before normal bladder control returned, we manually expressed the bladder of each rat once per day.

The second hemisection operation

The second hemisection operation was performed eight weeks after the first hemisection operation. Except for the vertebra receiving a laminectomy, other operation procedures were the same as in the first hemisection operation.

Basso, Beattie, Bresnahan (BBB) score tests in both hindlimbs

Motor performance was scored were performed according to the well-known Open Field BBB locomotor scale (Basso, Beattie & Bresnahan, 1995). It was given each week after each hemisection operation All of the BBB score tests, each lasting 5 min, were performed in an open field (diameter 150 cm) with a wood floor. When we monitored the movement of the hindlimbs, all rats moved freely without any disturbance. The paw placement, joint movements, weight bearing, and coordination among the limbs were used to evaluate the BBB locomotion scale.

Electrophysiological examinations

Before beginning electrophysiological examinations each rat was anesthetized. An electrophysiological examination was given before and after each hemisection operation.

Motor-evoked potential (MEP) studies on the body surface

MEP examination and analyses were performed mainly according to previously published methods (Shen et al., 2016; Yin et al., 2015; Ziegler et al., 2011). Before the examination, the rat was in a relaxed state. Intramuscular electrode needles were implanted in the anterior tibial muscle (TA) and little toe abductor muscle (LTA) on both sides. There were five wires used for MEP examination, and each wire was connected to a stainless steel pin. The rostral-caudal locations of the wires were as follows: the first needle was in the midline 5 mm from the nose; the second needle was located subcutaneously in the midline of the head; the third needle was located subcutaneously in the mid-belly; the fourth needle was located in the middle of the muscle being tests; and the fifth needle was located in the tail 3 cm from the root. Stimulations of 10 mA at 1 Hz were administered once per time point: for 1 ms per stimulus. Each muscle received three standard stimuli, and the interval time was 30 s.

Figure 3 Electrophysiological examinations of the spinal cord in first part of this study before and after the second hemisection operation at T7.

(A, B) Sham group, (C, D) Control group, (E, F) Ipsilateral group, (G, H) Contralateral group, (I, J) Transection group.

MEP on the spinal cord

The purpose of the test in spinal cord was to observe the change of conduction from T7–T8 after the operation. Eelectrophysiological examinations involved stimulating microelectrodes and recording microelectrodes (Fig. 3K) (Arvanian et al., 2009; Schnell et al., 2011). The responses evoked by stimulating the ventral horn from the rostral end of T7 to the caudal end of T8 on the ipsilateral side were recorded on the same side or on the contralateral side. The stimulation electrode was positioned approximately 0.7 mm from midline, with a depth of 1.3 mm, and an angle of 25–30° from the vertical sagittal plane. The recording electrodes were positioned approximately 0.7 mm from middle line, with a depth of 1.3 mm and an angle of 15–20° from the vertical sagittal plane. We used the average of two recordings for each side. There was a 30 s interval between the two stimuli. The ventral horn stimulus had a duration of 0.01 ms and a current of 0.5 mA and was delivered at 1 Hz.

Criteria for excluding animals

Rats were excluded from the research according to the following criteria: (1) death during or after the operation; (2) signs of autophagia and/or a serious skin infection; (3) an edematous hindlimb that would affect the BBB score test; (4) death during the electrophysiological recordings.

Statistics

The statistical analysis was performed using SPSS database (version 19.0; SPSS Inc., Chicago, IL, USA). The BBB scores and electrophysiological examination data are shown as means ±  SEM. When the data agreed with the Bartley Ball Test, the repeated measures general linear model test was used to determine the overall differences in the different test times after the operation (1 week to 6, 8, or 16 weeks), followed by LSD (least significant difference) tests to make comparisons among groups. P values less than 0.05 were considered statistically significant.

Results

Part 1: determination of an ipsilateral versus contralateral course

BBB score s after the first hemisection operation at T7

After the first operation, none of the rats could move their left hindlimb or perform weight-bearing movements, while the right hindlimb could move slightly. The most significant and rapid improvements in BBB scores of both hindlimbs occurred over the first two weeks. BBB scores continued to increase through the 3rd week after the operation and then reached a plateau phase in the 4th week that persisted through the end of the evaluation period at week 8 (Fig. 4A).

Figure 4 The BBB scores for part 1 of the study in which two hemisection operations were performed at T7.

A red arrow indicates the time point for the second hemisection. (A) The trend in BBB scores all rates of the first 8 weeks, before the second hemisection operation at T7. (B–E) Trends in BBB scores for each group after the first and second hemisection operations: (B) Sham group, (C) Control group, (D) Ipsilateral group, (E) Contralateral group, (F) Transection group. Data are presented as mean ± SEM.

BBB scores after a second hemisection operation at T7

A second hemisection at T7 had little effect on movement of either hindlimb in the ipsilateral group. In seven of eight rats BBB scores recovered to the pre-second operation level on the 3rd day and the BBB scores of all rats recovered to the pre-second operation level by the end of the first week following the second operation (Fig. 4D). No significant differences in the BBB scores between the ipsilateral group and the control group were observed after the operation (p < 0.05, Figs. 4C and 4D).

After the second hemisection at T7 on the contralateral side, movement in both hindlimbs was instantly obstructed and then started to recover two weeks later. By the 4th week, recovery entered a plateau phase. Compared to the rats in transection group, there were no significant differences in BBB scores of the ipsilateral group after the operation (p < 0.05, Figs. 4E and 4F). However, there were significant differences in BBB scores between the ipsilateral group and the contralateral group (p < 0.05, Figs. 4D and 4E).

Electrophysiological examinations in part 1 of the study

After the second hemisection at T7, the latency periods in both TA muscles and both LTA muscles between rats in the ipsilateral group and control group were not significantly different (p < 0.05, Figs. 5A–5D). However, they were longer in the contralateral group than in the control group (p < 0.05). There were no significant differences between the contralateral group and the transection group (p < 0.05). All experimental groups had significantly longer latency periods than the sham group (p < 0.05).

Figure 5 Electrophysiological examinations (MEP, latency periods) in body surface in part 1 part of this study before and after the second hemisection operation at T7.

(A–D) The latency periods in both TA muscles and both LTA muscles. *, P < 0.05, compared to the control group. #, P < 0.05, compared to the other 4 groups. Data are presented as mean ± SEM.

There were not significant differences in latency periods in the spinal cord between the sham, control, and ipsilateral groups (p < 0.05, Figs. 6A, 6B and 3A–3F). However, the latency period and wave amplitude disappeared in the contralateral and transection groups after the second hemisection (p < 0.05, Figs. 6A, 6B, 3G, 3H, 3I and 3J).

Figure 6 Electrophysiological examinations (MEP, latency periods) of the spinal cord in part 1 of this study, before and after the second hemisection operation at T7.

(A–B) The latency periods on both sides of the spinal cord before and after the second hemisection operation. Data are presented as mean ± SEM.

Part 2: determination of the involvement of vertebrae T8–T12

BBB scores after a second hemisection operation at T8–T12

In the T8 second hemisection group, BBB scores for the left hindlimb decreased slightly, while BBB scores of the right hindlimb were barely affected. Approximately three weeks later, BBB scores of both hindlimbs recovered to the level before the operation (p < 0.05, Fig. 7A). In comparisons of BBB scores six weeks after the second hemisection, the left hindlimbs in T8–T12 sub-groups displayed poorer and poorer improvement (p < 0.05, Figs. 7A–7E). Seven of eight rats in T12 operation sub-groups exhibited no left hindlimb movement 6 weeks after the operation.

Figure 7 Trends in BBB scores tendency for the T8 to T12 sub-groups in part 2 of the study.

(A–E) Operation group, (F–J) Control group, (K–O) Sham group. Data are presented as mean ± SEM.

Compared to the operation group, BBB scores for both hindlimbs in the T8–T12 sub-groups in the control group decreased significantly after the second hemisection (p < 0.05, Figs. 7A–7E and 7F–7J).

BBB scores of both hindlimbs in all T8–T11 sub-groups in the sham operation group recovered completely by the second week after the second hemisection (p < 0.05, Figs. 7K–7N). Compared to the T12 sub-group, the T8–T11 sub-groups displayed better improvement (p < 0.05, Figs. 7K–7O).

Electrophysiological examinations in part 2 of the study

After the second hemisection in the operation group, the latency period in left TA muscles and left LTA muscles became progressively longer from the T8 sub-group to T12 sub-group (p < 0.05, Figs. 8A and 8C).

Figure 8 Electrophysiological examinations (MEP, latency periods) of the T8–T12 sub-groups at the body surface in part 2 of this study after the second hemisection operation.

(A) Left TA muscle, (B) right TA muscle, (C) left LTA muscle, (D) right LTA muscle. *, P < 0.05, compared to T8 sub-group in each main group. #, P < 0.05, compared to the other four sub-groups. Data are presented as mean ± SEM.

Discussion

Effects of a second hemisection operation at T7

Many previous animal experiments have shown that after a hemisection operation of the spinal cord, the transplantation of stem cells or various engineered tissue materials in the injury region can improve movement of both hindlimbs. Immunohistochemical examinations have also shown that the nerve fibers in regions rostral to the injury site increased and deeply innervated the lesion site. This indicates that nerves fibers penetrating into the area of injury probably play important roles in recovery after spinal cord injury (Estrada et al., 2014; Fan et al., 2010; Jee et al., 2012; McCall, Weidner & Blesch, 2012; Wright et al., 2011).

However, some studies using animal models have reached conflicting conclusions. An important example is the work of Courtine et al. (2008) in which they carried out two successive lesions of the rat spinal cord and showed that the contralateral but not ipsilateral side was essential for recovery.

However, Courtine et al. (2008) did not performed a second hemisection at the injury region, so whether nerve regeneration within the injury region was attributable to hindlimb movements recovery could not be ruled out. Accordingly, in this study, we added an additional experimental group that received a second hemisection in the region of injury.

BBB scores for both hindlimbs decreased significantly after the first hemisection at T7, and 4 weeks later, the improvement reached a plateau. When the second hemisection operation was performed on the injury region, it had almost no effect on movement of either hindlimb (p < 0.05, Figs. 4C, 4D, 3C, 3D, 3E and 3F). The MEP results from body surface and spinal cord also displayed no significant differences (p < 0.05, Figs. 5A–5D, 6A, 6B, 3E, 3F, 3G and 3H). However, in the contralateral and transection groups, hindlimb movement on both sides disappeared after the second operation and there were no significant difference between groups at 8 weeks post-injury (p < 0.05, Figs. 4E and 4F). Moreover, MEP in the spinal cord disappeared after the second operation in both groups (Figs. 6A, 6B, 3G, 3H, 3I and 3J). By contrast, we found that, after the spinal cord transection operation MEP activities were still observed at the body surface. For this reason, MEP examination of the spinal cord is a more accurate indicator of injury and recovery than MEP at the body surface.

We thus conclude that the nerve repair in the injury region has no effect after the hemisection operation at T7 on the ipsilateral side, and reparative responses involved nerve fibers on the contralateral side.

Effects of a second hemisection operation at T8–T12 on the ipsilateral side

T8 subgroup

In T8 subgroup in operation group, the BBB scores of the left hindlimb decreased slightly, from the third week on, all rats returned to the level exhibited before the second operation (Fig. 7A). Compared to the control group, there were significant differences in the BBB score of both hindlimbs (p < 0.05, Figs. 7A and 7F). In the sham group, from the second week on, the sham operation almost had no effect to the movements of both hindlimbs. These results showed that the thoracic vertebra below and next to the spinal cord injury likely had a little assist in the recovery process, which may be affected by neuronal apoptosis around the injury site.

T9 subgroup

In T9 subgroup in operation group, the BBB scores of both hindlimbs decreased slightly and were approximately 1–2 points less than the level before the second hemisection in the 6th week (Fig. 7B). But the reduction of BBB scores was a little more than in the sham group (p < 0.05, Figs. 7B and 7L) and were less than in the control group (p < 0.05, Figs. 7B and 7G). These results showed that the spinal cord at thoracic vertebra T9 likely assisted with the recovery process to some extent.

T10 subgroup

In T10 subgroup in operation group, the BBB scores of left hindlimbs disappeared instantly and began at the 1st week (Fig. 7C). When compared to the T9 subgroup within the operation group, in the 6th week, the left hindlimb recovered were worse (Figs. 7B and 7C). This finding showed that the spinal cord at thoracic vertebra T10 played an important role in the spontaneous recovery of the spinal cord injury, and the function could not be completely compensated for by other segments of the spinal cord in six weeks.

T11 subgroup

In T11 subgroup in operation group, the BBB scores of left hindlimb disappeared instantly and began at the 2nd week (Fig. 7D). When compared to the T10 subgroup within the operation group, the left hindlimb recovered were worse and the difference was significantly in six weeks (Figs. 7C and 7D). This finding showed that the spinal cord at thoracic vertebra T11 played a very important role in the spontaneous recovery of the spinal cord injury.

T12 subgroup

In T12 subgroup in operation group, the BBB scores of left hindlimb disappeared instantly and were approximately 0–1 points from the 2nd to 6th week (Fig. 7E). These results showed that the spinal cord at thoracic vertebra T12 played a major role in the spontaneous recovery of the spinal cord injury.

In summary, results of hemisection at a single vertebra from T8–T12 after an initial hemisection at T7 impaired hindlimb movement recovery in all instances, with the most pronounced effects occurring at T11–T12. Therefore, we conclude based on the data in part 2 of this study that, when caudal to the injury region area, spinal cord segments underlying the T8–T12 vertebrae on the ipsilateral side all participated in the spontaneous recovery process after a hemisection operation at T7. In this case, the T12 vertebral area appeared to be the most important for nerve repair.

Repair processes occurring rostral to the lesion

Many findings have shown that, although the direct conduction pathway was destroyed in rats with a spinal cord injury, commands from brain conducted by the corticospinal tract (CST) can still be transmitted to the lumbar spinal cord below the lesion on the ipsilateral side (Bareyre et al., 2004; Courtine et al., 2008; Jankowska & Edgley, 2006; Kerschensteiner et al., 2004; Van den Brand et al., 2012). After injury, an important mechanism of the spontaneous recovery process is thus that the structure and course of nerve fibers in the CST are remodeled such that they make contact with propriospinal neurons that form detour pathways bypassing the lesion (Nishimura & Isa, 2012; Pierrot-Deseilligny, 2002; Rosenzweig et al., 2010; Zaaimi et al., 2012). However, the CST is not the only descending tract that affects movement and is not be the only projection system that conveys functional recovery (Han et al., 2013; Hurd, Weishaupt & Fouad, 2013). Spared reticulospinal fibers play an important role in the recovery process through spontaneous compensatory sprouting and increases in density after injury; they may also operate caudal to the lesion by enhancing indirect access to reticulospinal commands (Ballermann & Fouad, 2006; Zorner et al., 2014). Therefore, we hypothesize that the nerve fiber circuit underlying repair in this study was composed of CST and reticulospinal fibers and propriospinal neurons. We speculate that around the T12 vertebra, which was the most important for nerve repair, there were more nerve fibers relative to the other vertebrae that crossed the midline from the contralateral side to the injury side. However, more research, particularly nerve fiber tracing experiments, is needed to confirm this.

Repair processes occurring caudal to the lesion

Tillakaratne et al. (2010) showed that rats exhibited spontaneous recovery via a step-wise process after a complete transection of the mid-thoracic spinal cord, even though the region caudal to the spinal cord lesion did not make any connections to the brain and in absence of descending tracts passing through the lesion. Thus, roles of activity of a locally acting central pattern generator (CPG), which is present in many species, are important to consider (Deliagina et al., 1999; Ekeberg & Pearson, 2005; Grillner, 1985). While our research argues for the presence of significant spontaneous locomotor recovery resulting from new forms of dynamic control in the spinal CPG from newly generated or remodeled descending tracts, the local CPG still may play the primary role in recovery following spinal cord injury (Rossignol et al., 2007).

The CPG of the spinal cord is located in the lumbar enlargement at about the T1–12 segments (Magnuson et al., 1999). The second part of this study showed that areas closer to T12 vertebra are more important for nerve repair. Therefore, we could conceivably use various treatments to reinforce the role of CPG and the nerve fibers that connect to it to promote recovery. However, given that a second hemisection operation at T9 on the ipsilateral side could also impair hindlimb movements, which could not recover to pre-operation levels, the CPG is likely not the only factor playing a role in the recovery process.

Study limitations

These studies assessed spontaneous recovery from spinal cord injury, but not the impact of any treatment. Further research is needed to confirm if similar results are observed in the case of using a therapeutic intervention within the same experimental injury paradigm; potential therapies include targeting neuroinflammation, transplantation of engineered tissue materials, stem cells, neurotrophic factors, or genetic modifications. We also did not directly assess the path of newly projected nerve fibers to more conclusively and precisely define their course.

Conclusions

Our study demonstrates an anatomical mechanism for spontaneous repair processes caudal to spinal cord injury sites in which regenerative fibers cross to the contralateral side, course around the lesion, and then re-cross the midline innervating all caudal, ipsilateral vertebrae with T12 being most important. If we inject stem cells, neurotrophic factors, drugs in the spinal cord around the injury region more than only in the injury region, it might had more effect in the recovery process. Further studies should investigate therapeutic approaches that enhance this process and identify the molecular mechanisms that control it.

Supplemental Information

Supplemental Information 1 Electrophysiological examinations

Before beginning electrophysiological examinations each rat was anesthetized. An electrophysiological examination was given before and after each hemisection operation.

Click here for additional data file.

Supplemental Information 2 Basso, Beattie, Bresnahan (BBB) score tests in both hindlimbs

Motor performance was scored were performed according to the well-known Open Field BBB locomotor scale (Basso, Beattie & Bresnahan, 1995). It was given each week after each hemisection operation until the 8th week.

Click here for additional data file.

Additional Information and Declarations

Competing Interests

Author Contributions

Animal Ethics

Data Availability

The authors declare there are no competing interests.

Lu-sheng Li conceived and designed the experiments, performed the experiments, analyzed the data, contributed reagents/materials/analysis tools, wrote the paper, prepared figures and/or tables, reviewed drafts of the paper.

Hao Yu and Raynald Raynald performed the experiments, analyzed the data, contributed reagents/materials/analysis tools.

Xiao-dong Wang, Guang-hui Dai, Hong-bin Cheng and Xue-bin Liu performed the experiments, contributed reagents/materials/analysis tools.

Yi-hua An conceived and designed the experiments, contributed reagents/materials/analysis tools, reviewed drafts of the paper.

The following information was supplied relating to ethical approvals (i.e., approving body and any reference numbers):

Ethical approval was obtained from the Beijing Neurosurgical Institute Laboratory Animals Ethics Committee in China.

The following information was supplied regarding data availability:

The raw data has been supplied as Supplementary File.

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
