# Peer review of "Anatomical mechanism of spontaneous recovery in regions caudal to thoracic spinal cord injury lesions in rats"

_PeerJ, doi:10.7717/peerj.2865_

## Round 0.1 · original submission · Major Revisions

Dear Authors, The comments of these two reviewers are important to heed when you undertake major revisions to your manuscript.

Reviewer 1 ·

Basic reporting

This article presented with well written flow and fulfilled all the criteria of basic reporting inline with all PeerJ policies and standards. However, few minor suggestions that to be considered that may help the readers to understand better.

a) The hemisection surgery was performed according to Arvanian et al.,(2009). It is suggested if the authors explained briefly in few sentences in the text of what procedures that already did by cited paper. Please also mentioned if the authors have any modify techniques deviated from the cited paper. This will help the reader to get some ideas before reading the original article cited.
b) Basso Beattie Breshanan (BBB) locomotor tests were used in this study. However, the summary of how it performed was still not clear. It is suggested that the authors provide few sentences how it was performed. If it was performed by two blinded observers, what would be the variation between these two inter-observer variation.
c) This article proposed a very good surgery technique and none of the drop-out rate were reported. It is just a suggestion if the authors can provide some information of drop-out rate (by percentages% of dead animal throughout the study) and briefly explained the maintenance of the rats over 8 weeks of study period that would be great help for the readers or others to learn from this paper.

Experimental design

This research article proposed to revisit the spontaneous recovery of regenerated nerve after hemisected injury. This research article mainly reported the behavior performances of the both hindlimb (Basso Beattie Breshnan Locomotor Scale) and electrophysiology data (Motor evoked potential) to answer of the proposed valid questions. This is very interesting approach to investigate the re-innervation after hemisected injury by performing second hemisected surgery. However, several justification of the followings were needed to clarify what the authors proposed:

a) Why BBB score were read only for 6 weeks instead of 8 weeks for second group of hemisected surgery? Any justification for this will be a great help.
b) It is much appreciated if the authors can provide the information how the BBB were performed and if it is double blinded reading by two independent observers, the scores for second observer is needed.

The reviewer did not review on MEP data because it is out of his field of interest.

Validity of the findings

No comment

Additional comments

No comments

Reviewer 2 ·

Basic reporting

No Comments

Experimental design

No comments

Validity of the findings

No comments

Additional comments

The manuscript entitled “Anatomical mechanism of spontaneous recovery in regions caudal to thoracic spinal cord injury lesions in rats” was well written. The work is useful to understand the basic mechanisms of spontaneous recovery in spinal cord injury in animal model. Author used possible techniques to assess impact of lesions following spinal cord injury and findings are appreciable. Nevertheless, authors need to do some minor necessary corrections:

Author induced lesions in T7 and T8-12 in spinal cord, but why specifically used thoracic regions injury, did not write in introduction.

When you describe anatomical changes in spinal cord, in introduction, author must write elaborately spinal cord segments and associated with neuronal circuits.

Doses of anesthesia did not indicate. Also, did author use any kind of analgesic and antibiotics? It would be helpful for researchers.

Methods and results were explained well. Results presentation is appreciable.

Legends are too many sentences, better to reduce.

In discussion section, author again slightly described results. So discussion is too short. Need to discuss findings still more clear. The present study is not completed as shown in the limitations of study, however, author may suggest some valuable drugs to treat or speed up process, it would be appreciable.

In discussion, author should describe about his own perspective for spontaneous recovery in molecular mechanisms. It would be interesting.

---

## Round 0.2 · accepted · Accept

Dear Authors,

Thank you for the recent submission of your revised manuscript which has been accepted by the peer reviewers.

Reviewer 1 ·

Basic reporting

No comments

Experimental design

No comments

Validity of the findings

No comments

Additional comments

This manuscript was revised and well written according to the reviewers suggestions. I hope the readers will get something useful information from this paper.

Reviewer 2 ·

Basic reporting

No Comments

Experimental design

No Comments

Validity of the findings

No Comments

Additional comments

Author did all necessary changes. I recommend to accept the current form of manuscript.